# Effect of Fluoride Varnishes on Demineralization and Acid Resistance in Subsurface Demineralized Lesion Models

**DOI:** 10.3390/jfb15120380

**Published:** 2024-12-17

**Authors:** Rika Iwawaki, Taku Horie, Abdulaziz Alhotan, Yuka Nagatsuka, Keiko Sakuma, Kumiko Yoshihara, Akimasa Tsujimoto

**Affiliations:** 1Department of Operative Dentistry, School of Dentistry, Aichi Gakuin University, Nagoya 464-8651, Japan; iwawaki@dpc.agu.ac.jp (R.I.); lifedoor@dpc.agu.ac.jp (T.H.); yuca@dpc.agu.ac.jp (Y.N.); virgo@dpc.agu.ac.jp (K.S.); 2Department of Oral Health Sciences, BIOMAT, KU Leuven, 3000 Leuven, Belgium; 3Department of Dental Health, College of Applied Medical Sciences, King Saud University, P.O. Box 10219, Riyadh 12372, Saudi Arabia; aalhotan@ksu.edu.sa; 4Health and Medical Research Institute, National Institute of Advanced Industrial Science and Technology (AIST), Takamatsu 761-0395, Japan; kumiko.yoshihara@aist.go.jp; 5Department of Pathology & Experimental Medicine, Graduate School of Medicine, Dentistry and Pharmaceutical Sciences, Okayama University, Okayama 700-8558, Japan; 6Department of Operative Dentistry, University of Iowa College of Dentistry, Iowa City, IA 52242, USA; 7Department of General Dentistry, Creighton University School of Dentistry, Omaha, NE 68102, USA

**Keywords:** dental biomaterials, surface analysis, enamel, fluoride, remineralization

## Abstract

This study aimed to clarify the effects of high-concentration fluoride varnish application on the inhibition of the progression of initial enamel caries. Remineralization capacity and acid resistance following high-concentration fluoride varnish application were compared with untreated models and models treated with fluoride mouthwash. Bovine enamel was used to create a model of initial enamel caries. The high-concentration fluoride varnishes Enamelast and Clinpro White Varnish and the fluoride mouthwash Miranol were used. Specimens were evaluated using Contact Microradiography (CMR) and an Electron Probe Micro-Analyzer (EPMA). While a single application of high-concentration fluoride varnish and short-term fluoride mouthwash use did not appear to cause remineralization in the subsurface demineralized layer, improvements in acid resistance were observed, leading to reduced demineralization under subsequent acidic challenges.

## 1. Introduction

Dental caries are among the most prevalent infectious diseases in the world, affecting over 90% of adults according to a survey conducted by the US CDC from 2011 to 2016 [1]. This prevalence has not changed substantially since an earlier survey in 1999–2004. Caries are strongly influenced by the production of organic acids by cariogenic bacteria, and these acids demineralize the tooth structure. In initial enamel caries, subsurface demineralized lesions are formed while the superficial mineralized layers are retained, and these are clinically observed as white spots [2]. Generally, in the enamel surface, decalcification and recalcification occur repeatedly, and when this balance is disturbed and demineralization becomes predominant, the subsurface demineralized lesions progress and eventually develop into caries [3,4]. The concept of Minimal Intervention Dentistry (MID), originally proposed by the World Dental Federation (FDI) in 2002, and then modified in 2016, recommends the improvement of oral microflora through plaque control and sugar restriction, patient education through guidance on diet and oral hygiene, and remineralization therapy for such initial enamel caries [5]. The ICDAS (International Caries Detection & Assessment System), which was proposed in 2005, classifies caries in detail according to observable features and classifies these initial enamel caries lesions as code I or II, recommending remineralization treatment and ongoing observation [6]. It is now broadly accepted that initial enamel caries can be treated through remineralization and that a surgical approach is not necessary except in cases of high esthetic requirements.

One approach to treating such initial enamel caries includes the use of fluoride varnish. These materials are easy to apply to the teeth and include colorings and flavorings that ensure that they are pleasant for the patient. They remain in place on the tooth for at least 24 h, even in thin layers, and are reported to continue releasing fluoride over that period. This material has been approved for treating dental hypersensitivity in the USA and Japan, and it is widely used “off-label” for caries prevention and to promote remineralization [7]. Fluoride varnishes have been used in Europe for sixty years, and the evidence for their safety is extremely strong [8]. Furthermore, clinical studies have shown that the application of these materials at routine check-ups can reduce the incidence of caries by between 25% and 45% in patients ranging from young children to adults [8]. In the UK, such treatment is available through the National Health Service up to twice a year, regardless of age. The effectiveness and safety of this approach are widely recognized in Europe, as shown by evidence from several Cochrane reviews [8]. This treatment is believed to promote both remineralization and acid resistance. However, although the epidemiological evidence for its effectiveness is very strong, understanding of its mechanism of action is still inadequate.

This study uses a laboratory initial enamel caries model to evaluate two possible mechanisms for the anticariogenic effect of this treatment: the remineralization ability of fluoride varnish, and the changes in acid resistance of the area to which it is applied.

## 2. Materials and Methods

### 2.1. Preparation of Enamel Specimens

Three specimens per tooth were cut from the labial surface of freshly extracted bovine anterior teeth (bought from Kenis Limited (Osaka, Japan), https://global.kenis.co.jp) using a low-speed cutting machine (Isomet, Buehler, Lake Bluff, IL, USA), which were 4 mm wide in the direction of the tooth axis, 5 mm long perpendicular to the tooth axis, and 4 mm thick. The specimens were then polished to #2000 under water using silicon carbide paper so that the labial enamel surfaces were as flat as possible. The pulpal surface of the specimen was then removed parallel to the enamel surface so that the final thickness of the specimen was 3 mm. The specimens were then ultrasonically cleaned in distilled water for 1 min and masked with nail varnish to create a 3 × 4 mm rectangular surface in the center of the labial enamel, which was used as the treatment surface (Figure 1).

### 2.2. Subsurface Demineralization Model

Specimens with a subsurface demineralized layer were prepared for different remineralization treatments. The preparation for making specimens with a subsurface demineralized layer was carried out using lactic acid following the methods of Hayashi [9]. The surface was immersed in 20 mL (37 °C) of 8% Methocel MC gel (Fluka, Buchs, Switzerland) for 24 h, with the treated surface facing upwards, and then a sheet of filter paper of approximately the same size as the inner diameter of the container was laid over the surface of the gel and 20 mL of 0.1 mol/L lactic acid solution (pH 4.6) was gently applied to the paper. The specimens were demineralized for 10 days to make the subsurface demineralized model (Figure 2).

### 2.3. Fluoride Treatment

The subsurface demineralized models were randomly divided into four groups. The first group (E group) was treated with Enamelast (Ultradent Products, South Jordan, UT, USA) according to the manufacturer’s instructions. The second group (W group) was treated with Clinpro White Varnish (Solventum, St. Paul, MN, USA), again according to the manufacturer’s instructions. The third group (M group) was immersed in Miranol (Bee Brand Medico Dental, Osaka, Japan) for ten minutes twice a day for a period of three days. The final group was left untreated as a control group. All specimens were stored in artificial saliva (Saliveht Aerosol, Teijin Pharma, Tokyo, Japan) for 3 days, except when they were being treated. The surfaces of these specimens were finally cleaned with distilled water.

### 2.4. Re-Demineralization

The fluoride-treated samples were immersed into an artificial demineralizing solution (50 mmol/L acetic acid, 1.5 mmol/L CaCl_2_, 0.9 mmol/L KH_2_PO_4_, pH 4.6, 37 °C) for 3 days [9].

### 2.5. CMR and EPMA Analysis

All prepared samples of each group of specimens were dehydrated with an ethanol system and embedded in epoxy resin (Polysciences, Warrington, PA, USA). A thin section of each specimen was prepared for Contact Microradiogram (CMR) analysis through extraction of the central part of the specimen parallel to the tooth axis using a low-speed cutting machine. The bulk sections remaining after the extraction of the thin section were used for EPMA analysis. The sections extracted for CMR were prepared with a thickness of approximately 300 µm, and they were refined into 100 µm thick sections by finishing with #2000 SiC paper. The sectioned surfaces of the other remaining blocks were also polished up to #2000 SiC paper for Electron Probe Micro-Analyzer (EPMA) analysis (Figure 3).

### 2.6. CMR Imaging

Polished sections of each group were subjected to CMR photography using a soft X-ray non-destructive inspection system with an acceleration voltage of 10 kV, a tube current of 2 mA, and an irradiation time of 5 min, avoiding drying of the sections as much as possible, together with aluminum step wedges for calibration curves. A band at the center of the section approximately 100 µm wide was imaged over a distance of 250 µm from the enamel surface edge of the specimen, representing a depth of 250 µm from the surface of the treated sample. A glass dry plate (High Precision Photo Plate, Konica Minolta, Tokyo, Japan) was used for imaging, and a special developing solution (CDH-100, Konica Minolta) and a fixing solution (CFL-881, Konica Minolta) were used for developing and fixing, respectively.

### 2.7. Mineral Profiling

The CMR of each group was digitized using a camera (DP70, Olympus, Tokyo, Japan) connected to an optical microscope (SZX9, Olympus), and a mineral profile was created using image analysis software (WinROOF Ver3.4,0, Mitani Corporation, Fukui, Japan). The gray values were measured in 256 shades of gray. The obtained gray values were corrected with reference to the gray value of the corresponding aluminum step wedge.

The gray value of the area on the film where the tooth sample was not present was taken to represent 0 vol% of mineral, and the gray value of the area where sound enamel was visible was taken to represent 100 vol% of mineral. These values were used to calculate the demineralization at each depth of the sample, and the amount of mineral loss (ΔZ, vol%·µm) was calculated. In other words, the amount of mineral loss was calculated from the area between the graph for one condition and another. The mineral losses (ΔZ) between fluoride and re-demineralization models were measured.

### 2.8. EPMA Observation

The specimens for EPMA analysis of each group were mirror polished with 0.3 µm alumina oxide and ultrasonically cleaned in distilled water for 1 min. The surface was then treated with carbon deposition, and the microstructure of the longitudinal section was observed using a field emission electron probe micro-analyzer (FE-EPMA, JXA 8530F, Jeol, Tokyo, Japan) at an acceleration voltage of 10 kV. The wavelength range of detected X-rays was 0.087 to 9.3 nm, with a 3 nm resolution for the secondary electron imaging.

### 2.9. Elemental Analysis

In order to investigate the distribution of minerals on the surface where the microstructure was observed, a surface analysis (acceleration voltage 10 kV, acceleration current 3.0 × 1r8A) was carried out for Ca and P using FE-EPMA.

## 3. Results

### CMR and Mineral Profiles

Figure 4 and Figure 5 show the typical CMR and mineral profiles for each of the various fluoride treatment models.

Figure 6 shows the typical EPMA analysis for each of the various fluoride treatment models.

In the model of enamel subsurface demineralization, the typical profile of such demineralization was observed, with mineral loss just below the outermost layer of enamel, although the outermost layer of enamel was retained. The mineral density was lowest around a depth of about 30 μm in the subsurface demineralized layer, and the mineral density at that depth was 4.0 vol%.

Using these mineral profiles as a reference, we defined the area from the outermost layer to a mineral density of 50% in the demineralized layer model, which extended from the outermost layer to a depth of around 60 μm, as the “sub-surface demineralised layer”, and the area from a density of 50% to the intact part, which extended from a depth of around 60 μm to around 100 μm, as the “deep sub-surface demineralised layer”.

In the fluoride application model, the CMR and mineral profiles of the M, W, and E groups were the same as those of the enamel subsurface demineralization model.

On the other hand, in the re-demineralization model, different CMR and mineral profiles were observed depending on the type of fluoride treatment. In the control group, although the outermost layer of enamel was retained, the CMR of the deep subsurface demineralized layer was observed to be significantly demineralized, and mineral loss was observed in the mineral profile to a depth of 50 μm below this layer.

In groups M, W, and E, the outermost layer of enamel and the deep subsurface demineralized layer were preserved in the CMR, but transparency was observed in the lower part of the subsurface demineralized layer. On the other hand, the trends in the mineral profiles of groups M, W, and E were different.

In the M group, the mineral density of the subsurface demineralized layer was maintained, but a decrease in mineral density was observed in the layer below it, with the greatest loss of mineral density occurring at a depth of around 130 μm, where the mineral density was 77.3 vol%.

In the mineral profiles of groups W and E, a decrease in mineral density in the subsurface demineralized layer was observed, but the amount of mineral loss was smaller than in the control group. On the other hand, a marked decrease in mineral density was observed in the lower layer, with the greatest mineral density loss occurring at a depth of around 100 μm in the W group and around 120 μm in the E group, with mineral densities of 15.5 vol% and 55.9 vol%, respectively.

The mineral loss (vol %·μm) for the control, E, W, and M groups in these representative cases was 7410.0, 3353.5, 5589.5, and 554.3, respectively.

In the EPMA analysis, similarly to mineral changes observed in CMR, loss of Ca and P was identified in the subsurface enamel layer. The distributions of Ca and P were almost completely identical.

## 4. Discussion

It has been widely reported that the application of fluoride to demineralized lesions in the enamel surface layer promotes remineralization [10]. However, in the present experiment, there was no recovery of mineralization or increase in Ca and P concentrations in the subsurface demineralized layer in any of the E, W, or M groups in the fluoride application model. Previous studies have reported that for fluoride to promote remineralization, a continuous supply of calcium and fluoride ions contained in saliva is necessary. Therefore, it was thought unlikely that remineralization of the subsurface demineralized layer would be promoted by fluoride application over the short period of time (3 days) covered in this study. In addition, when fluoride is applied to the subsurface demineralized layer of the enamel, the fluoride ions may replace hydroxide ions in the enamel and remain in the superficial layer. Therefore, although immersion in artificial saliva (mineral solution) for a long period of time may promote remineralization through reactions with the calcium and phosphate ions contained in it, and there is a possibility that remineralization will be observed under these conditions, further investigation is needed.

The condition of the demineralized enamel samples that were treated with fluoride and then re-demineralized showed different trends in each of the E, W, and M groups compared to the control group. It is generally known that the application of fluoride improves resistance to acid in healthy enamel as well as in the subsurface demineralized layer. In this study, too, the M, W, and E groups formed a layer that showed resistance to acid in the lower part of the subsurface demineralized layer, and the deeper part of the sample was selectively demineralized, unlike the control groups. Hayashi et al. reported that when the subsurface demineralized layer of enamel that had undergone remineralization treatment was demineralized again, the surface layer and the subsurface demineralized layer acquired acid resistance, and the deeper layer below was more strongly attacked by the acid [9]. The results of this study are consistent with previous reports, which argue that fluorapatite, which is resistant to acid, is formed through the replacement of hydroxide ions in demineralized hydroxyapatite with fluoride ions [11]. The reason for demineralization directly beneath the layer with acid resistance was thought to be that demineralization starts in the deeper layers of enamel that were not reached by the fluoride, and so that part is selectively demineralized.

Turning to CMR imaging, there was no clear difference between the images of the subsurface demineralized samples from the control group and from the treatment groups. Fluorine (F) is generally considered to have low X-ray contrast, so even if fluoride compounds accumulated on the surface of the demineralized area after fluoride application, it is likely difficult to observe this using CMR. On the other hand, when the demineralized layer was re-demineralized, the elution of calcium ions was suppressed in the layer where fluorapatite was formed, and it was expected that this would cause a difference in the amount of minerals measured using CMR, as was observed.

When comparing the effects of different fluoride treatments on inhibiting demineralization, the results of this study showed that the mineral loss in groups W (Clinpro Varnish) and E (Enamelast) was greater than in group M (Miranol), despite the fact that the fluoride concentration of the fluoride varnishes, Clinpro Varnish and Enamelast, was higher than that of the fluoride mouthwash, Miranol, at 22,600 ppm. Miranol, a mouthwash, is a solution containing a low concentration of fluoride (250 ppm F), and it can be thought that the fact that it was more easily able to penetrate into the demineralized area under the enamel surface made it more effective than the high-concentration, but high-viscosity, fluoride varnishes in suppressing mineral loss. According to a previous report on the amount of fluoride ions released from fluoride varnish, the amount of fluoride released when Clinpro Varnish was immersed in water for 24 h was 74.0 ± 32.2 μmol/g [12]. This suggests that only about 1/16 of the fluoride, which is present at a high concentration of 22,600 ppm in the material itself, was actually released. Additionally, as this fluoride is diluted in the surrounding water, the concentration of ions available to react with the demineralized layer is likely to be even lower. On the other hand, although the fluoride concentration in Miranol is low at 250 ppm, it is all in aqueous solution, suggesting that the ion concentration available to react with the demineralized layer may be higher in the mouthwash than in the varnish.

When fluoride treatment is applied to the subsurface demineralized layer, the acquisition of an acid-resistant layer and remineralization are also influenced by contact with the mineral components contained in saliva, so, in this experiment, the specimens were immersed in artificial saliva in order to more accurately reproduce the intraoral environment. In other words, the mineral components contained in saliva are essential for the acquisition of an acid-resistant layer and remineralization of the enamel through a chemical reaction with fluoride ions [13]. Bolis et al. reported that the amount of fluoride uptake by enamel from a high-concentration fluoride varnish applied directly to it does not necessarily correlate with the amount of fluoride released in solution [14]. In addition, Kim et al. reported that the improvement in surface hardness was greater when fluoride varnish was applied indirectly to the enamel compared to when it was applied directly, suggesting that fluoride varnish may prevent calcium and phosphate ions in saliva from penetrating into demineralized areas by covering the enamel [15]. In other words, fluoride varnish includes substances, such as rosin, that secure both adhesiveness and insolubility, and although it can remain in the area where it is applied for a long time, it may be that the increase in fluoride concentration in the surrounding area is limited and that the layer of varnish may also reduce the supply of other ions to that area. Godoi et al. studied the effect of fluoride varnishes on remineralization, using hardness as a proxy for mineral content, and they also found no remineralization of the subsurface lesion [16].

In this study, we conducted experiments in the hope that a single application of high-concentration fluoride varnish would result in high acid resistance and remineralization, but no significant remineralization was observed in either group, and the creation of acid-resistant layers was less effective than that with fluoride-containing mouthwash. However, it is worth noting that a significant improvement in acid resistance was observed in the group that received a single short application of fluoride varnish compared to the untreated group. Because fluoride mouthwashes require continuous application by the patients themselves, it is possible that a single application of fluoride varnish may be an effective way to secure some increase in acid resistance in patients who may not cooperate with using a mouthwash, such as young children, or who have difficulty with rinsing, such as the elderly.

## 5. Conclusions

The application of high-concentration fluoride varnish or fluoride mouthwash to an early enamel caries model was found to not significantly promote remineralization in the subsurface demineralized layer, but it improved the acid resistance of the deepest part of the layer.

## Figures and Tables

**Figure 1 jfb-15-00380-f001:**
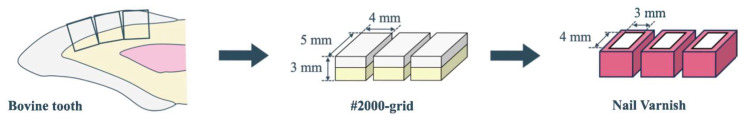
The preparation of enamel specimens.

**Figure 2 jfb-15-00380-f002:**
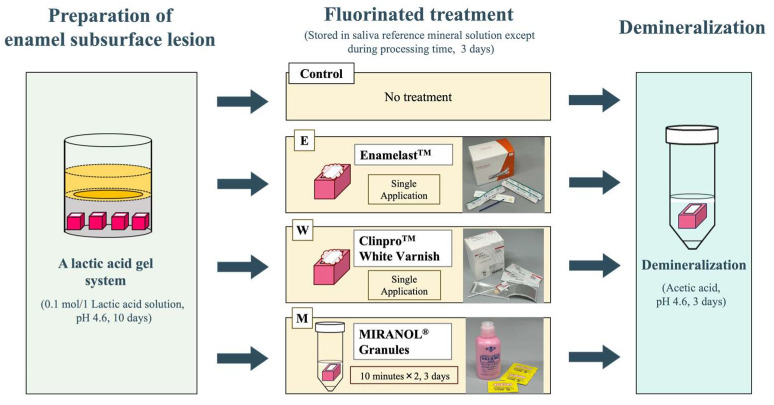
The procedures for preparation of enamel subsurface lesion, fluorinated treatment, and demineralization.

**Figure 3 jfb-15-00380-f003:**
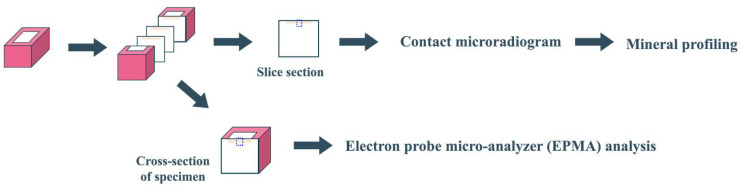
The observation and analysis.

**Figure 4 jfb-15-00380-f004:**
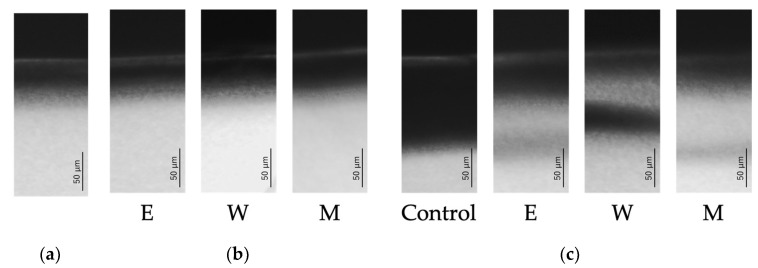
The typical CMRs. (**a**) Prepared enamel subsurface lesion. (**b**) After fluorinated treatment. (**c**) After demineralization.

**Figure 5 jfb-15-00380-f005:**
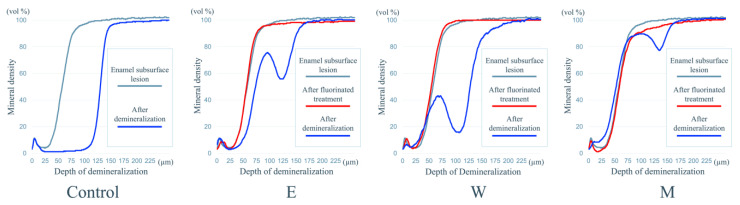
Typical mineral profiles.

**Figure 6 jfb-15-00380-f006:**
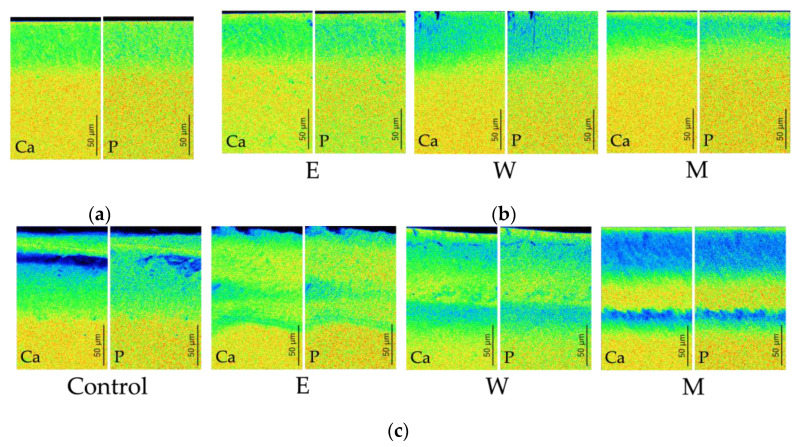
Typical EPMA analysis. (**a**) Prepared enamel subsurface lesion. (**b**) After fluorinated treatment. (**c**) After demineralization.

## Data Availability

The original contributions presented in the study are included in the article, further inquiries can be directed to the corresponding author.

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
