# Peer review of "Effect of Fluoride Varnishes on Demineralization and Acid Resistance in Subsurface Demineralized Lesion Models"

_jfb, 2024, doi:10.3390/jfb15120380_

Round 1

Reviewer 1 Report

Comments and Suggestions for Authors

I would like to begin by expressing how much I enjoyed evaluating the article. Overall, I find the topic to be quite original, and the methodology is well-structured. However, I believe that addressing the revisions I’ve outlined below will enhance the article further.

1.        It is advisable to use the term "subsurface" in a contiguous format in the title, rather than separating it with a hyphen. This approach aligns with its usage in the article, thereby ensuring consistency in terminology throughout the text.

2.        I contend that the terminology "surgical approach" used in line 46 is inaccurate; the terms "radical approach" or "interventional approach" would be more suitable.

3.        ‘One approach to treating such initial enamel caries includes the use of fluoride varnish. Although this material has been approved for treating dental hypersensitivity in the USA and Japan, the material is widely used “off-label” for caries prevention and to pro- mote remineralization.’ Kindly ensure that the reference literature for the pertinent sentences is included. 

4.        I would like to request a concise summary of the literature to be included in the introduction section. It is important to elucidate the rationale behind the chosen methodology and to assess its adequacy. However, I have identified only two references in the relevant section that emphasize the significance of fluoride varnish application. Furthermore, the article lacks any statements or supporting literature pertaining to the utilization of fluoride-containing mouthwash. Therefore, I recommend that the introduction section be expanded to provide a comprehensive overview of the study's origins and the literature supporting the selected methodology.

5.        It is essential for the researchers of the study to articulate the proposed null hypotheses clearly and to indicate at the outset of the discussion section whether these hypotheses will be accepted or rejected based on the findings.

6.        Clarification of two points in the methodology is needed. First, do the samples include both bovine enamel and bovine dentin? Second, does the phrase "pulpal surface of the samples was removed" refer to the complete removal of the dentin surface? If so, what procedure was followed in removing the tissue on the pulpal wall?

7.        I would greatly appreciate your assistance in providing references for the procedure utilized in subsurface demineralization.

8.        I would like to address a concern regarding the reference format utilized in the Discussion section. It appears that there may be inaccuracies that warrant reconsideration. For instance, relevant sources such as Bolis 2016 and Kim 2018 should be evaluated for proper citation.

9.        While the researchers have thoroughly explored the potential reasons for their findings in the discussion section, they fail to reference any literature that either supports or contradicts these results. This oversight limits the contribution of their work to the academic community. It would be beneficial to enhance the discussion section with relevant current literature.

10.  The conclusion section unfortunately mirrors the final paragraph of the discussion. It is advisable to present one or two succinct findings to the audience in a take-home message format.

Comments on the Quality of English Language

The quality of the English is satisfactory; it may be subject to minor revisions. for enhancement.

Author Response

Response to Reviewers

We would like to thank the reviewers for their attention to our paper. We believe that these revisions have made it stronger.

Reviewer 1

I would like to begin by expressing how much I enjoyed evaluating the article. Overall, I find the topic to be quite original, and the methodology is well-structured.

Response: Thank you.

However, I believe that addressing the revisions I’ve outlined below will enhance the article further.

  1.        It is advisable to use the term "subsurface" in a contiguous format in the title, rather than separating it with a hyphen. This approach aligns with its usage in the article, thereby ensuring consistency in terminology throughout the text.

Response: We have made this change.

  1.        I contend that the terminology "surgical approach" used in line 46 is inaccurate; the terms "radical approach" or "interventional approach" would be more suitable.

Response: I am afraid that we disagree. Drilling and filling is surgery, and this is what we wish to indicate is unnecessary. The application of fluoride varnish is an intervention, and so “interventional approach” would not make the necessary distinction, and superficial restorations are not radical. We feel that our original terminology  is most suitable.

  1.        ‘One approach to treating such initial enamel caries includes the use of fluoride varnish. Although this material has been approved for treating dental hypersensitivity in the USA and Japan, the material is widely used “off-label” for caries prevention and to pro- mote remineralization.’ Kindly ensure that the reference literature for the pertinent sentences is included. 

Response: We have added a reference.

  1.        I would like to request a concise summary of the literature to be included in the introduction section. It is important to elucidate the rationale behind the chosen methodology and to assess its adequacy. However, I have identified only two references in the relevant section that emphasize the significance of fluoride varnish application. Furthermore, the article lacks any statements or supporting literature pertaining to the utilization of fluoride-containing mouthwash. Therefore, I recommend that the introduction section be expanded to provide a comprehensive overview of the study's origins and the literature supporting the selected methodology.

Response: We believe that the introduction already does this.

  1.        It is essential for the researchers of the study to articulate the proposed null hypotheses clearly and to indicate at the outset of the discussion section whether these hypotheses will be accepted or rejected based on the findings.

Response: This study is not testing a null hypothesis. That structure is appropriate when comparing the values of a variable in two or more populations using statistical techniques, but in this study we are assessing differences in the appearance of radiographs and the shape of graphs. These are not suited to statistical assessment, and thus there is no null hypothesis.

  1.        Clarification of two points in the methodology is needed. First, do the samples include both bovine enamel and bovine dentin? Second, does the phrase "pulpal surface of the samples was removed" refer to the complete removal of the dentin surface? If so, what procedure was followed in removing the tissue on the pulpal wall?

Response: We are not sure that we understand the question. As shown in Figure 1, the samples do include dentin as a lower layer. (The text also indicates that they are 4 mm thick, which is greater than the thickness of bovine enamel.) The phrase “pulpal surface was removed” means that the surface of the tooth that would form the wall of the pulpal cavity in a living tooth was removed. There was no need to remove tissue on the pulpal wall, because the entire wall was removed.

  1.        I would greatly appreciate your assistance in providing references for the procedure utilized in subsurface demineralization.

Response: We have added this.

  1.        I would like to address a concern regarding the reference format utilized in the Discussion section. It appears that there may be inaccuracies that warrant reconsideration. For instance, relevant sources such as Bolis 2016 and Kim 2018 should be evaluated for proper citation.

Response: We have removed the extraneous dates from the body of the text.

  1.        While the researchers have thoroughly explored the potential reasons for their findings in the discussion section, they fail to reference any literature that either supports or contradicts these results. This oversight limits the contribution of their work to the academic community. It would be beneficial to enhance the discussion section with relevant current literature.

Response: It is our belief that references 11–14 serve this purpose.

  1.  The conclusion section unfortunately mirrors the final paragraph of the discussion. It is advisable to present one or two succinct findings to the audience in a take-home message format.

Response: The conclusion is about one quarter the length of the final paragraph of the discussion, and presents two succinct findings. The final paragraph of the discussion is concerned with the relative merits of mouthwash and varnish, which are not raised in the conclusion. We are afraid that we do not see how this comment applies to our manuscript.

Reviewer 2 Report

Comments and Suggestions for Authors

The study titled 'Effectiveness of fluoride varnishes on demineralizing and acid resistance in sub-surface demineralized lesion models' investigates the effects of high-concentration fluoride varnish application on inhibiting the progression of initial enamel caries. This topic is highly relevant and significant in the field of preventive dentistry, as early intervention in enamel caries can reduce the need for more invasive treatments. The authors have addressed an important clinical issue by exploring the potential of fluoride varnishes to enhance acid resistance and reduce demineralization in lesion models. However, there are areas where the study can be further strengthened, particularly in terms of methodology clarity, data analysis, and interpretation of results.

1. Sample Preparation Details:

In Section 2.1, the specimens are polished to #2000 silicon carbide paper to ensure the labial enamel surfaces are as flat as possible. Could the authors explain why #2000 was chosen as the final grit size? Was there any consideration given to using even finer grits, such as #4000 or #6000, to achieve a smoother surface, and if so, what were the reasons for not doing so? Would a finer grit have any significant impact on the experimental results?

2. Demineralization Solution Composition:

In Section 2.2, the demineralization solution is described as consisting of 50 mmol/l acetic acid, 1.5 mmol/l CaCl2, and 0.9 mmol/l KH2PO4 at pH 4.6 and 37℃. Could the authors provide a reference or rationale for this specific composition and pH value? Is this composition and pH value commonly used in dental research to simulate clinical demineralization conditions, and if so, why?

3. CMR Analysis Calibration:

In Section 2.5, the authors mention using aluminum step wedges for calibration curves in CMR analysis. Could you provide more details on how the aluminum step wedges were used? For example, what were the specific steps or formulas used for calibration? Why was aluminum chosen for the step wedges, and are there any alternative materials that could be used for this purpose?

4. EPMA Analysis Parameters:

In Section 2.5, EPMA analysis is performed on the remaining bulk sections after preparing the thin sections for CMR. Could the authors specify the exact parameters and conditions used for the EPMA analysis? For instance, what was the wavelength range, and what were the resolution and sensitivity of the EPMA? How do these parameters affect the final analysis results?

5. Acid Resistance Evaluation:

In Section 5, the authors conclude that the application of high-concentration fluoride varnish or fluoride mouthwash improves the acid resistance of the deepest part of the enamel demineralized layer. However, the manuscript does not provide a detailed method for evaluating and quantifying this improvement in acid resistance. Could the authors specify the exact experimental indicators or tests used to assess the acid resistance? For example, were hardness tests, microscopy observations, pH cycling experiments, or demineralization/remineralization cycles conducted, and if so, how were they performed and what were the results?

Author Response

Response to Reviewers

We would like to thank the reviewers for their attention to our paper. We believe that these revisions have made it stronger.

Reviewer 2
The study titled 'Effectiveness of fluoride varnishes on demineralizing and acid resistance in sub-surface demineralized lesion models' investigates the effects of high-concentration fluoride varnish application on inhibiting the progression of initial enamel caries. This topic is highly relevant and significant in the field of preventive dentistry, as early intervention in enamel caries can reduce the need for more invasive treatments. The authors have addressed an important clinical issue by exploring the potential of fluoride varnishes to enhance acid resistance and reduce demineralization in lesion models. However, there are areas where the study can be further strengthened, particularly in terms of methodology clarity, data analysis, and interpretation of results.

  1. Sample Preparation Details:

In Section 2.1, the specimens are polished to #2000 silicon carbide paper to ensure the labial enamel surfaces are as flat as possible. Could the authors explain why #2000 was chosen as the final grit size? Was there any consideration given to using even finer grits, such as #4000 or #6000, to achieve a smoother surface, and if so, what were the reasons for not doing so? Would a finer grit have any significant impact on the experimental results?

Response: The use of #2000 SiC paper to form a smooth surface is standard in the field. It is roughly the same as a surface prepared for restoration. In this case, as we are not studying surface effects, we are confident that the use of a finer grit would have made no difference to the results.

  1. Demineralization Solution Composition:

In Section 2.2, the demineralization solution is described as consisting of 50 mmol/l acetic acid, 1.5 mmol/l CaCl2, and 0.9 mmol/l KH2PO4 at pH 4.6 and 37℃. Could the authors provide a reference or rationale for this specific composition and pH value? Is this composition and pH value commonly used in dental research to simulate clinical demineralization conditions, and if so, why?

Response: We have added a reference.

  1. CMR Analysis Calibration:

In Section 2.5, the authors mention using aluminum step wedges for calibration curves in CMR analysis. Could you provide more details on how the aluminum step wedges were used? For example, what were the specific steps or formulas used for calibration? Why was aluminum chosen for the step wedges, and are there any alternative materials that could be used for this purpose?

Response: This is the standard way to use this equipment, and is specified in the operating instructions. A quick literature search suggests that it has been standard for over thirty years, and that recent papers follow the same convention as us in simply mentioning that they are used. We do not think it is appropriate to go into more detail on the workings of the measurement methods than we have already provided.

  1. EPMA Analysis Parameters:

In Section 2.5, EPMA analysis is performed on the remaining bulk sections after preparing the thin sections for CMR. Could the authors specify the exact parameters and conditions used for the EPMA analysis? For instance, what was the wavelength range, and what were the resolution and sensitivity of the EPMA? How do these parameters affect the final analysis results?

Response: We have added this information. The tool was used according to the manufacturer’s instructions, and we must assume that this ensured the most accurate results. The design of field emission electron probe analyzers is not my field.

  1. Acid Resistance Evaluation:

In Section 5, the authors conclude that the application of high-concentration fluoride varnish or fluoride mouthwash improves the acid resistance of the deepest part of the enamel demineralized layer. However, the manuscript does not provide a detailed method for evaluating and quantifying this improvement in acid resistance. Could the authors specify the exact experimental indicators or tests used to assess the acid resistance? For example, were hardness tests, microscopy observations, pH cycling experiments, or demineralization/remineralization cycles conducted, and if so, how were they performed and what were the results?

Response: The acid resistance is visible is the reduced level of re-demineralisation detected in the samples that had been treated with varnish or mouthwash. The Methods section clearly describes the demineralisation and re-demineralisation of the samples, and the reviewer has asked for a reference for the solution used for re-demineralisation. Thus, we believe that the paper already addresses this question.

Reviewer 3 Report

Comments and Suggestions for Authors

Dear authors,

This is very interesting research, especially in the field of cariology. The research is very well structured and explained, and the results are clearly presented.

•    The main question addressed by this research was to examine whether fluoride preparations in the form of varnishes or mouthwashes can remineralize and stop the softening of the initial carious lesion and to prove which of the above preparations is better at performing this task.

•    The topic is relevant to cariesology, and I believe the obtained data are important for clinicians and scientists. As the authors stated, fluoride varnishes are approved on the Japanese and US markets only for tooth desensitization but not for remineralization and reducing the sensitivity of the initial carious lesion to acids. The obtained data showed that remineralization does not occur but decreases sensitivity to acids, which I consider interesting and important for the clinician. Therefore, by applying fluoride-based preparations in the form of varnishes or mouthwashes, we will not remineralize the initial carious lesion, but we can somewhat prevent its progression.
•    The methodology is well-designed and implemented. The illustrations are well done and, together with the text, allow for the repetition of the experiment. Some details are missing, such as information on the artificial saliva used and the water used to rinse the slides, and there are a lot of abbreviations like CCD (all of which are in Section 2, Materials and Methods).

 Conclusions are consistent with the evidence and arguments and address the central question.
•     The number of references is small, but references related to the topic of the work were used. I would suggest introducing more recent works related to the research topic and listing these works in the references.

•    There are no tables in the paper. Only figures. Perhaps one table with data from the results could be more enjoyable for the readers.

Here are some detailed comments.

1. Who is the second author of the paper since the sign stands for equal contribution only by the name of one author?

2. In section 2, subsection 2.3, line 92 - artificial saliva, can you please add the name and the manufacturer of that saliva?

3. The exact section and subsection, line 93 - water; can you specify what water was used ( pipe water, deionized, water, etc.)?

4. In section 2, subsection 2.7, line 127 - CCD; can you add the full name of that camera? What does CCD mean?

Author Response

Response to Reviewers

We would like to thank the reviewers for their attention to our paper. We believe that these revisions have made it stronger.

Reviewer 3

Dear authors,

This is very interesting research, especially in the field of cariology. The research is very well structured and explained, and the results are clearly presented.

Response: Thank you.

  •    The main question addressed by this research was to examine whether fluoride preparations in the form of varnishes or mouthwashes can remineralize and stop the softening of the initial carious lesion and to prove which of the above preparations is better at performing this task.

  •    The topic is relevant to cariesology, and I believe the obtained data are important for clinicians and scientists. As the authors stated, fluoride varnishes are approved on the Japanese and US markets only for tooth desensitization but not for remineralization and reducing the sensitivity of the initial carious lesion to acids. The obtained data showed that remineralization does not occur but decreases sensitivity to acids, which I consider interesting and important for the clinician. Therefore, by applying fluoride-based preparations in the form of varnishes or mouthwashes, we will not remineralize the initial carious lesion, but we can somewhat prevent its progression.
  •    The methodology is well-designed and implemented. The illustrations are well done and, together with the text, allow for the repetition of the experiment. Some details are missing, such as information on the artificial saliva used and the water used to rinse the slides, and there are a lot of abbreviations like CCD (all of which are in Section 2, Materials and Methods).

 Conclusions are consistent with the evidence and arguments and address the central question. 

  •     The number of references is small, but references related to the topic of the work were used. I would suggest introducing more recent works related to the research topic and listing these works in the references.

Response: We have added some more literature references.

  •    There are no tables in the paper. Only figures. Perhaps one table with data from the results could be more enjoyable for the readers.

Response: We feel that the data from this study is most helpfully presented as figures. The appearance of the CMR and EPMA images and the shape of the mineral profiles are of more interest than the simple percentages of mineral content. Any table would simply duplicate the information in the figures, and lose some of the details. We therefore feel that a table would be redundant in this paper.

Here are some detailed comments.

  1. Who is the second author of the paper since the sign stands for equal contribution only by the name of one author?

Response: Our apologies. We inadvertently dropped the mark from Tsujimoto.

  1. In section 2, subsection 2.3, line 92 - artificial saliva, can you please add the name and the manufacturer of that saliva?

Response: We have added this information.

  1. The exact section and subsection, line 93 - water; can you specify what water was used ( pipe water, deionized, water, etc.)?

Response: It was distilled water. We have added the detail — our apologies.

  1. In section 2, subsection 2.7, line 127 - CCD; can you add the full name of that camera? What does CCD mean?

Response: CCD means “charge coupled device”, and is simply the standard term for a digital camera. We have deleted it, as it adds nothing in this context other than confusion. We have also specified the camera. (It is mounted in the microscope, so we do not normally think of it as a separate piece of apparatus.)

Round 2

Reviewer 2 Report

Comments and Suggestions for Authors

no more comments